# Technical note: Comparing three different methods for allocating river points to coarse-resolution hydrological modelling grid cells

Juliette Godet[1], Eric Gaume[1], Pierre Javelle[2], Pierre Nicolle[1], and Olivier Payrastre[1]

[1]GERS, Univ. Gustave Eiffel, IFSTTAR, 44344 Bouguenais, France
[2]RECOVER, INRAE, Université d'Aix-Marseille, 13100 Aix-en-Provence, France

**Correspondence:** Juliette Godet (juliette.godet@univ-eiffel.fr)

**Abstract.** The allocation of points in a river network to pixels of a coarse-resolution hydrological modelling grid is a well-known issue, especially for hydrologists who use measurements at gauging stations to calibrate and validate distributed hydrological models. To address this issue, the traditional approach involves examining grid cells surrounding the considered river point and selecting the best candidate, based on distance and upstream drainage area as decision criteria. However, recent studies have suggested that focusing on basin boundaries rather than basin areas could prevent many allocation errors, even though the performance gain is rarely assessed. This paper compares different allocation methods and examines their relative performance. Three methods representing various families of methods have been designed: area-based, topology-based and contour-based methods. These methods are implemented to allocate 2580 river points to a 1km hydrological modelling grid. These points are distributed along the entire hydrographic network of the French southeastern Mediterranean region, covering upstream drainage areas ranging from $5km^2$ to $3000km^2$. The results indicate that the differences between the methods can be significant, especially for small upstream catchments areas.

## 1 Introduction

In hydrology, rainfall-runoff models' outputs are often compared to observed discharge series at gauging stations for calibration or evaluation purposes. Vector-based hydrological models are adequate to meet these objectives, because it is straightforward to locate a gauging station along the river network. However, when using gridded models, it is necessary to allocate each gauging station to a specific cell in the model grid. In the literature, terms such as "co-registering" (Fekete et al., 2002), "co-referencing" (Döll and Lehner, 2002), and "matching" (Wang et al., 2018) are also used to describe this process. The allocation of specific river points to a coarse-resolution cell can also be necessary when connecting the output of a hydrological distributed model (providing hydrographs or peak discharges on a grid) to a hydraulic model for inundation modelling. As an example, Dottori et al. (2017) developed a European-wide flood risk assessment system, based on the European Flood Awareness System (EFAS, see Thielen et al. 2009; Bartholmes et al. 2009). The discharge output of EFAS is provided on a grid of spatial resolution of $5km$, which needed to be downscaled to a resolution of $100m$ in order to derive flood hazard maps at the pan-European scale. Dottori et al. (2015) opted for a basic method consisting in allocating the $100m$ river pixel to the $5km$ river cell containing

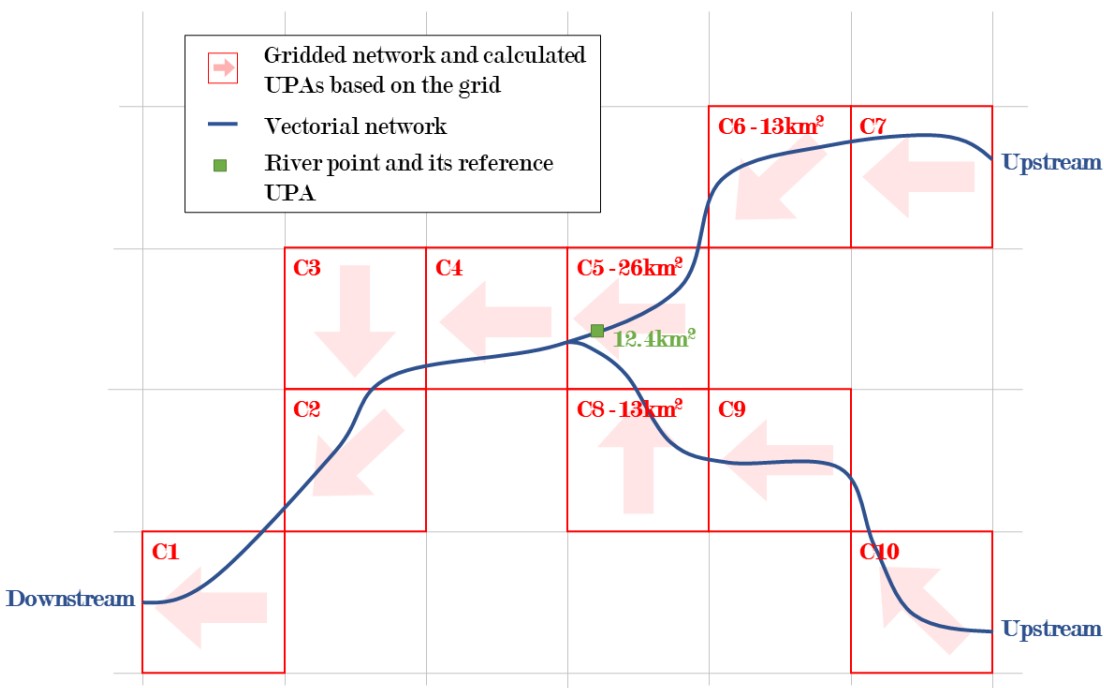

**Figure 1.** Example of allocation process failure when based on distance and UPA error criteria: green river point is allocated to grid cell C8 instead of grid cell C5.

this pixel. This approach has limitations, particularly when the two river networks defined at the $100m$ and $5km$ scales do not overlap.

Generally, the allocation of river points to a coarse-resolution grid for hydrological modelling relies on distance and upstream drainage area (UPA) error criteria (Döll and Lehner, 2002; Fekete et al., 2002; Lehner, 2012; Zhao et al., 2017; Sutanudjaja et al., 2018; Wang et al., 2018; Burek et al., 2020; Polcher et al., 2022). However, this process is also prone to errors, especially near confluences, where points of different branches of the river network may have similar UPAs. As a result, their allocation on the coarse-resolution grid can lead to assigning a point to the wrong hydrological grid cell and corresponding upstream watershed, based on a slightly better UPA fit or a slightly shorter distance (see figure 1 for an example).

Considering this possible limitation, efforts have been made to propose more effective protocols for allocating river points to hydrological grid cells. For instance, the methods proposed by Burek and Smilovic (2022); Munier and Decharme (2022) combine distance and UPA error criteria with a comparison between the gauging station's basin boundaries, delineated on the basis of a fine-resolution Digital Elevation Model (DEM), and the basin boundaries of the allocated cell based on the coarse-resolution hydrological grid. In both studies, the similarity between the watershed limits is characterized by the Intersection Over Union index (Rezatofighi et al., 2019).

The idea of comparing basin boundaries had been previously considered, although in a slightly different context, namely the evaluation of hydrological grids obtained from an upscaling algorithm (i.e., transforming a fine-resolution grid into a coarser-resolution grid). Initially, upscaling algorithms were also guided by a comparison between the UPA values of "small pixels" and the corresponding upscaled "large cells" (Reed, 2003; Paz et al., 2006; Yamazaki et al., 2008; Eilander et al., 2021). Visual inspections were performed to identify obvious inconsistencies between small scale and upscaled grids and corresponding river networks. To reduce these inconsistencies automatically, additional criteria have been proposed to complement the UPA criterion for the optimisation of uspcaling algorithms such as the mean distance between river networks and the percentage within a buffer (Davies and Bell, 2009), the correctness index and the figure of merit (Li and Wong, 2010), and the watershed delineation percentages of consistency (Sousa and Paz, 2017).

In summary, numerous methods are available to achieve the objective of allocating a river point to a coarse-resolution grid cell. However, these methods have been developed in different contexts and have rarely been compared. This study aims at comparing the results obtained from three different types of methods for allocating a large number (2580) of river points to a coarse-resolution hydrological grid ($1km \times 1km$). The first method belongs to the category of area-based methods and employs distance and UPA error criteria. The third method is a contour-based approach. The second method is a topological method based on proximity along the river network. Another unique aspect of this work is that it deals with a detailed river network that includes river points with small drainage areas (minimum of $5km^2$), whereas most previous studies have been limited to the main river networks (catchments larger than $500km^2$ in Dottori et al. (2017) for instance). In this study, $1km \times 1km$ is considered as "coarse" resolution because the hydrological model is intended for the regional scale. However, the same problem could arise for hydrological modelling applied on a continental scale where the resolution will be coarser than 1km (i.e. 5 to 10 km).

The paper is structured as follows: Section 2 presents the three tested allocation methods as well as the validation metrics. Section 3 provides an overview of the case-study. Finally, Section 4 compares and discusses the results obtained with the three tested allocation methods.

## 2 Allocation methods and validation metrics

In this section, we describe three methods that allow for the allocation of a river point to a coarse-resolution grid cell. To implement and evaluate these methods, it is necessary to have reference catchment boundaries for each river point, which can be obtained from a fine-resolution Digital Elevation Model (DEM).

### 2.1 Method 1: area-based method

Area-based methods can be traced back to Döll and Lehner (2002), who proposed allocating river points to the coarse-resolution grid cells containing the points, provided that the relative difference between coarse and reference resolution UPAs did not exceed 5%. This criterion led, in their case, to a manual re-allocation of 35% of the points. In order to automate the allocation procedure, Lehner (2012) proposed to select the grid cell within a 5km radius search area around each river point (see Figure

2) with the lowest value of a discrepancy criterion $D = RA + 2R$, where $RA$ stands for the relative difference between UPAs that should not exceed 50% and $R$ for the distance between the point and the centre of the grid cell. In most other works (Zhao et al., 2017; Sutanudjaja et al., 2018; Wang et al., 2018; Burek et al., 2020), RA is the only selection criterion, the radius of the search area (1-25km, depending on the spatial resolution) and the maximum acceptable RA value for a successful allocation (10-30%) varying between studies. To maximise the proportion of allocated river points and to optimise the computation time,

the approach proposed herein, proceeds in three possible successive steps. At step 1, the closest grid cell verifying $RA < 10\%$ and $R < 3$ cells is selected, if it exists. If it does not, the maximum RA values is increased to 20% at step 2 and 30% at step 3. It can be noted that the proposed approach combines area and distance criteria. Also, if a maximum difference between UPAs of 30% is a recurrent choice in the literature, e.g (Burek et al., 2020), regardless of the studied model resolutions, the distance criterium R<3 is more study-dependant. In the present study, it appeared after some tests as a good compromise providing accurate results with reasonable computation times. However when using global-scale hydrological models and coarser grids,

accurate results with reasonable computation times. However when using global-scale hydrological models and coarser grids, the value of R may have to be adjusted.

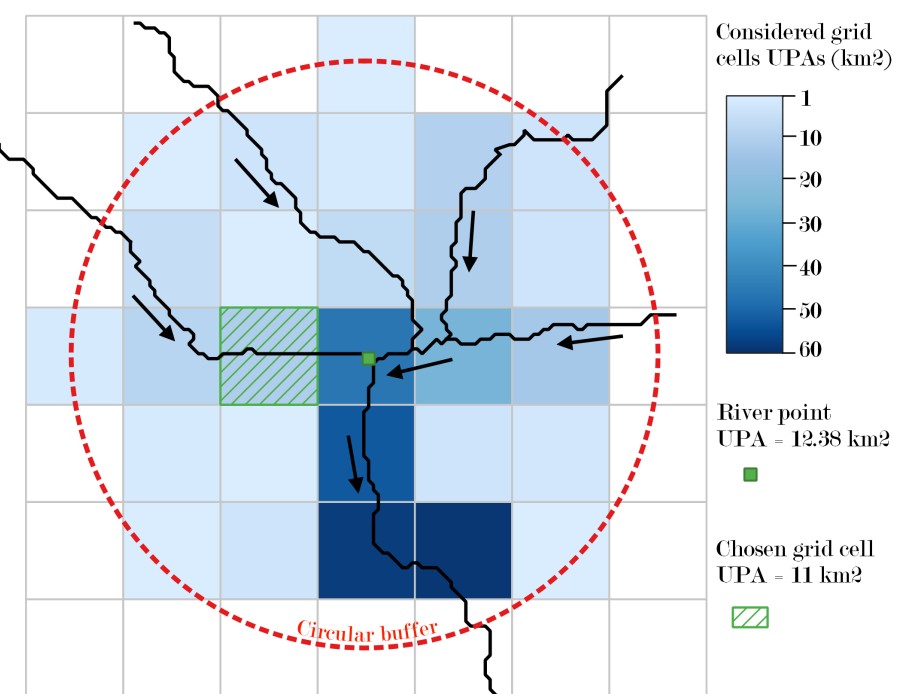

**Figure 2.** Illustration of Method 1: grid cells candidates for a specific river point. In this situation, the river point is allocated to the green-hatched grid cell.

.

## 2.2 Method 2: topology-based method

This second method requires a vector-based river network and the definition of coarse grid cells' *outlet points*. The cells' outlet points are located and selected according to the IHU upscaling method (Eilander et al., 2021), used to generate the coarse-resolution hydrological modelling grid (see section 3.2). Each river point can then be connected to the closest upstream or downstream grid cell outlet point and hence allocated to the corresponding grid cell, provided that both points belong to the same river reach, i.e. are not separated by a network confluence (case of point P3 in Figure 3). With this method, river points located between two confluences within the same grid cell cannot be allocated.

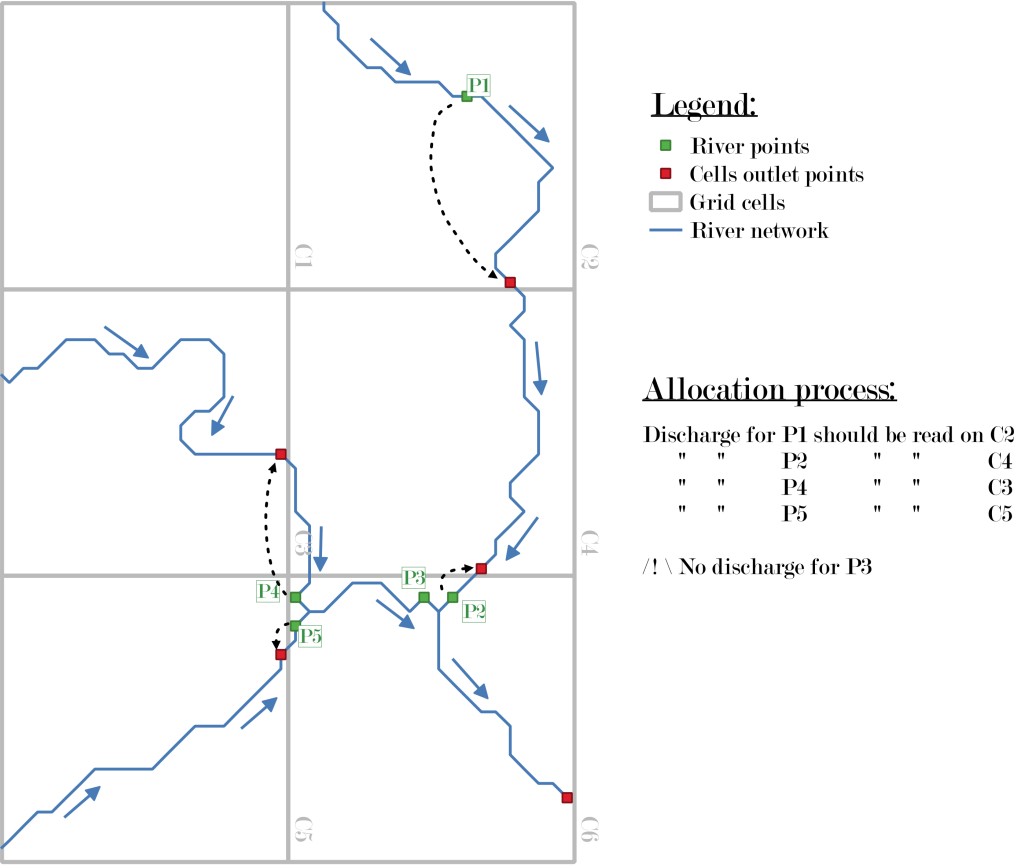

**Figure 3.** Illustration and limits of Method 2: connection of river points and cells outlet points (black dotted arrows) with an impossibility for point P3, located between two confluences in the same grid cell.

## 2.3 Method 3: contour-based method

Several previous works have stressed the importance of considering the consistency of watershed contours for the evaluation or optimisation of upscaling or allocation methods (Davies and Bell, 2009; Li and Wong, 2010; Sousa and Paz, 2017). Method 3 is similar to method 1, except that it is not only based on the comparison between upstream watershed surfaces, but also between upstream watershed contours. While Munier and Decharme (2022); Burek and Smilovic (2022) did propose an allocation criterion based on a combination of an area-based and a contour-based criterion, it is proposed herein to base the selection of the appropriate grid cell for each river point based on a single criterion, namely the critical success index, CSI (see eq. 1). The CSI is a standard score to compare surfaces, often used to compare flood inundation models for instance (Fleischmann et al., 2019; Hocini et al., 2021).

$$CSI = \frac{a}{a+b+c} \tag{1}$$

Where $a$ (HIT, see figure 4) is the overlapping area between the reference upstream watershed of the considered river point and the upstream watershed of the candidate grid cell, defined on the coarse-resolution grid using the TAUDEM library (Tarboton, 1997). "b" is the area of the reference watershed not overlapping with the coarse grid watershed (MISS) and conversely, "c" is the area of the coarse grid watershed, not overlapping with the reference watershed (FALSE ALARMS). $CSI$ is equal to 1 in case of a perfect overlap and 0 when there is no overlap at all. It can be noted that the CSI has often been used with alternative denominations in previous studies, such as the Intersection over Union criterion (Munier and Decharme, 2022; Burek and Smilovic, 2022), the Figure of Merit (Li and Wong, 2010), or Fit Metric (Fleischmann et al., 2019).

Like for method 2, the allocation procedure proceeds in two possible steps. The grid cell maximising the CSI value is selected among the 9 cells closest to the considered river point. If a minimum CSI value is not reached (i.e. 0.4 for watershed areas under 10 $km^2$ and 0.6 otherwise), the search area is extended to the $7 \times 7 = 49$ closest grid cells. Tests carried out prior to this choice showed that extending the research area to the $5 \times 5 = 25$ closest grid cells did not allow most of the targeted cases to be corrected. The CSI thresholds can be adjusted (see section 4.4), as well as the surface threshold and the extended research area, since they may depend on the model resolution.

## 2.4 Evaluation metrics

The allocation procedures aims at relating points of the river network, often corresponding to a gauging station, to coarse grid cells of a distributed hydrological model, with the upstream watershed closest to the actual watershed, or at least closest to the watershed delineated based on the finest available geographical data. Therefore, the CSI appears as the best suited score for the efficiency evaluation of allocation methods and will be used hereafter. The CSI is calculated based on the low resolution catchment contours (reference contours). By construction, method 3 should then lead to the best performances, but at the cost of a higher implementation complexity and significantly larger computation times, as will be illustrated herein. Hence, the main question will be : "how well do methods 1 and 2 compared to method 3 ?"

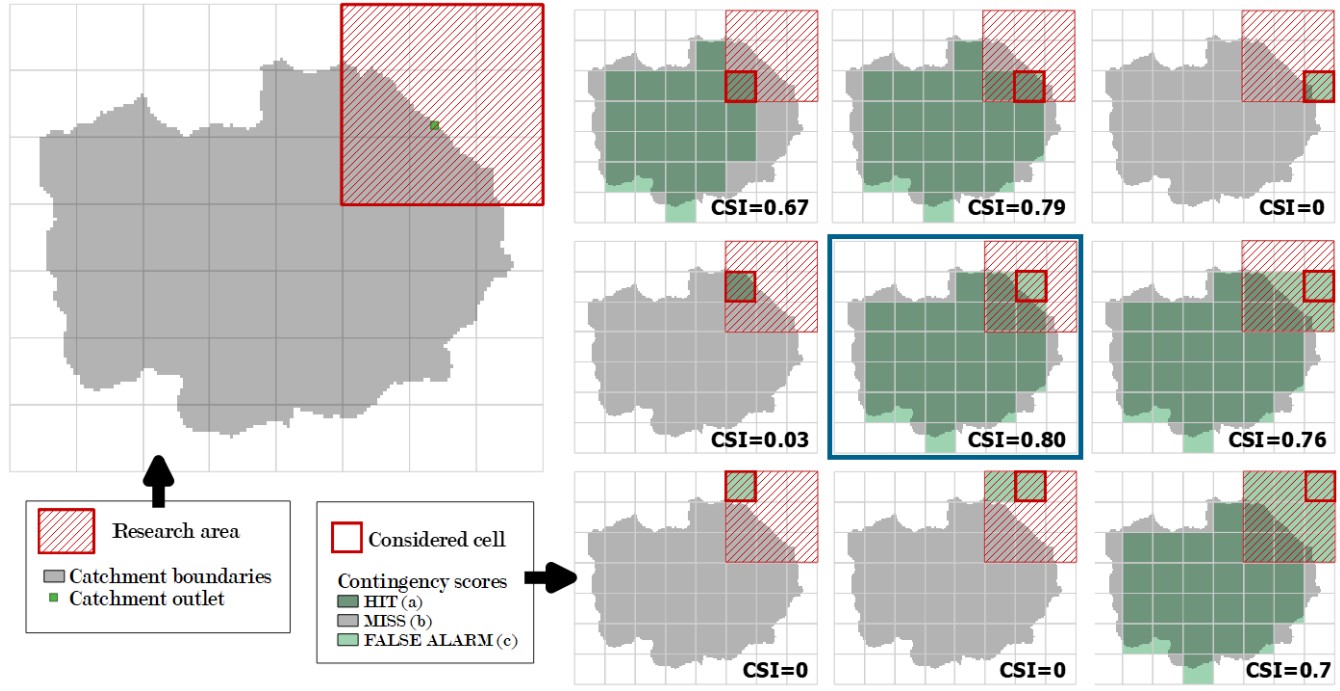

**Figure 4.** Illustration of Method 3: CSI calculations for the nine candidate cells. In this situation, the river point is allocated to the candidate cell of the blue box scenario, which maximises the CSI (CSI = 0.80).

.

## 3    Case study and data

The test area covers three departments in the Eastern Mediterranean region of France, with a total surface area exceeding 15,000 $km^2$ (figure 5). Two geographical datasets have been used to implement and compare the allocation methods: the river points to be allocated and their reference catchments boundaries, and the coarse-resolution grid specifically designed for the purpose of the study.

### 3.1    The river points to be allocated

The BNBV (Base Nationale des Bassins Versants) is a French reference GIS layer describing the network of rivers with an upstream catchment area larger than $5km^2$, over the whole territory of France. It was produced by Organde et al. (2013), based on the processing of a hydrologically-validated 50m-resolution flow direction grid. It includes a vector description of the river reaches and identifies approximately 15,000 points of interest along the river network across mainland France. These points correspond to locations of gauging stations, urban areas, confluences, river mouths. Intermediate outlets were also added to ensure comprehensive coverage of the whole river network. The upstream watershed limits and areas are associated to each BNBV point.

Figure 5 displays the BNBV outlets and river reaches in the Eastern Mediterranean region. BNBV outlets located in the étang de Berre area were excluded from our study due to the lack of meaningful flow direction in a lagoon. The region contains 2580 BNBV outlets that will be allocated to a coarse-resolution grid. The vector representation of the upstream catchment limits serves as the reference for the evaluation and for the implementation of Method 3, while the vector description of river reaches is essential for Method 2.

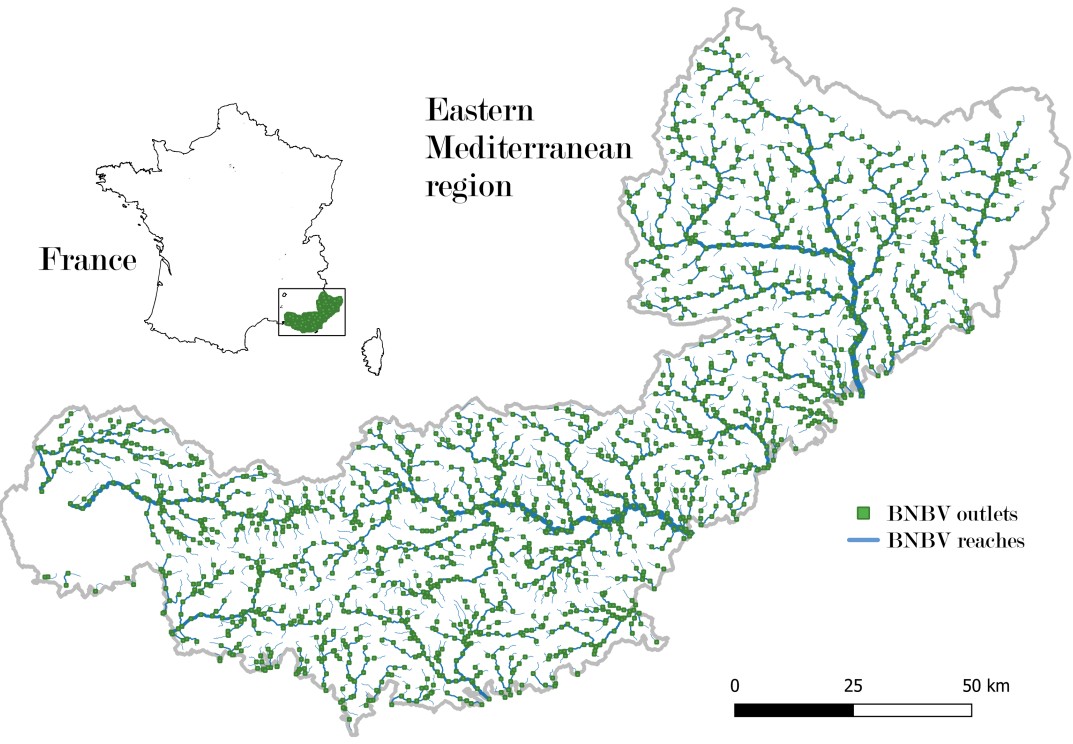

**Figure 5.** The 2580 BNBV outlets on the French Eastern Mediterranean region.

### 3.2 The coarse-resolution hydrological modelling grid

Regional gridded hydrological models are often implemented on 1km resolution grids, aligning with the typical resolution of operational radar-based quantitative precipitation estimates. In this context, the $1km \times 1km$ hydrological modelling grid was herein generated by upscaling the 50m flow direction grid from the BNBV database using the IHU method (Eilander et al., 2021), see figure 6 to visualize the upscaling results. The IHU method incorporates principles from previous upscaling methods (Döll and Lehner, 2002; Fekete et al., 2001; Olivera et al., 2002; Paz et al., 2006; Wu et al., 2011) and has demonstrated superior performance compared to the benchmarks methods, e.g. DMM method (Olivera et al., 2002) and EAM method (Yamazaki et al., 2008). Moreover, the IHU method is the only fully automated and open-source flow direction grid upscaling method known to the authors. After implementing the IHU method, we made minimal manual corrections to the flow direction grid. Only a

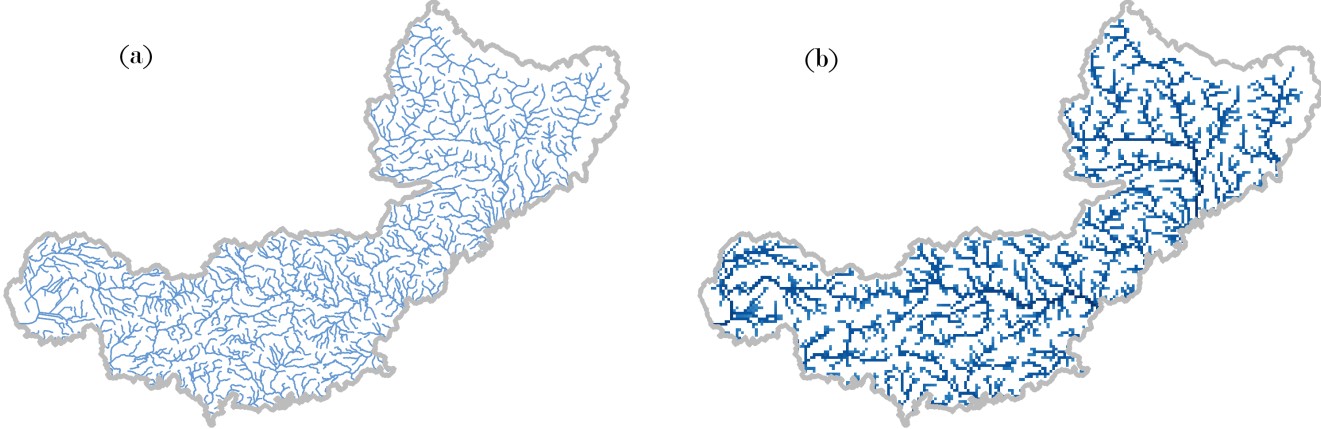

**Figure 6.** Initial $50m$ river network vectorized for $S > 5km^2$ (a) and $1km$ upscaled surface accumulation ($S > 5km^2$) grid (b).

small number of cells, approximately a dozen out of around 14,000 cells, required adjustments. These manual corrections were primarily made along the zone's borders, particularly near the coastline.

## 4  Results

### 4.1  Advantages and disadvantages of each method

Table 1 provides an overview of the general advantages and disadvantages of each method. Method 1 and 2 demonstrate superior performance in terms of computation times. The most time-consuming stage of Method 3 is the computation of the upstream catchment for each candidate cell, and its comparison with the reference catchment. The duration of these processes will naturally increase with the size of the considered case study and the number of points to be allocated. However, the computation times might not be considered as a crucial issue, since the allocation of river points to grid cells need do be performed only once, before hydrological modelling. Method replicability is probably a more important issue for this work. Method 2 is limited in its compatibility as it relies on the IHU upscaling, As a consequence, it makes this method inoperable in contexts where the coarse resolution gridded network does not coincide with the river network (i.e. both networks may come from different data sources). On the other hand, Method 3 requires reference catchment boundaries. Furthermore, the confidence level varies between the methods. Method 1 relies solely on basin area information, which means that near confluences, it may allocate a river point to a neighboring grid cell with a similar basin area but belonging to a different catchment. Method 2 ensures that the river points and chosen grid cell belong to the same river reach, while Method 3 selects the cell with the most similar upstream catchment in terms of basin contour and location. Lastly, Method 3 guarantees the allocation of all river points, covering 100% of the dataset, whereas the other two methods may not achieve complete allocation.

| | Method 1 | Method 2 | Method 3 |
|---|---|---|---|
| **Computation times** | $\simeq$ Second | $\simeq$ Minute | $\simeq$ Hour |
| **Replicability** | Replicable | Requires the definition of *cells outlet points* | Replicable |
| **Additional data** | None | River network vector and cell outlet points | Reference basin limits |
| **Confidence level** | Variable | High | High |
| **% of allocated outlets** | Potentially $< 100$ | Potentially $< 100$ | 100 |

**Table 1.** Advantages and disadvantages of each method.

It is also important to note that river points similar to P3 in Figure 3, i.e. points located between two confluences within the same grid cell, are not allocated with Method 2. These points represent about 2% of the total number of points to be allocated in the considered region. One potential solution could consist in selecting, among the closest upstream or downstream outlets of grids, the one with the nearest usptream watershed area. Another solution would be to allocate P3 to both upstream cells (C3 and C5 in Figure 3), however in this paper we chose not to permit the allocation to several grid cells. In these cases, Method

2 detects that all allocations to a single grid cell will be imperfect, which is why it was not deemed essential to propose an allocation solution for these specific points, for the sake of comparison.

### 4.2   Comparison of allocation performances

In order to compare the quality of allocation among the three methods, we initially examined the CSI statistics. However, it should be noted that while Method 1 and Method 3 successfully allocated all 2580 considered river points (BNBV outlets in

the Eastern Mediterranean zone), Method 2 only allocated 2532 points (98%). To ensure a fair comparison, these 48 points were excluded from the analysis. Figure 7 displays the histograms of CSI values for each method, along with the corresponding mean and median values. The results indicate that Method 1 has significantly lower performances compared to Methods 2 and 3. This discrepancy is primarily attributed to a high percentage of river points with very low CSI scores (less than 0.05) in Method 1. These low scores occur when a river point is allocated to a cell solely based on similar basin area, disregarding

substantial differences in the shape and location of the upstream catchments (e.g. figure 1).

     In its state, Method 2 cannot allocate 100% of the river outlets, even though it could with a small extension (see previous section). Method 3 was able to allocate the excluded outlets successfully, with a mean CSI of 0.7. Additionally, Figure 7 highlights that Method 3 consistently yields a minimum CSI of around 0.25, whereas Method 2 falls below 0.05 (with 12 river points having CSI < 0.25). Figure 8 provides a more detailed comparison of CSI scores between Method 2 and Method 3. It

clearly demonstrates the consistent superiority of Method 3 over Method 2. While the CSI differentials are generally small, the green circle in the figure highlights cases where the differential can be significant. These are characterised by river points with small upstream basin areas (less than $12km^2$) located far from the nearest cell outlet point, resulting in notable differences in UPAs. Figure 9 shows an example of these situations, which are often complex configurations where different branches of the network are in close proximity to each other, often within the same grid cell. As a consequence, the upscaled flow directions

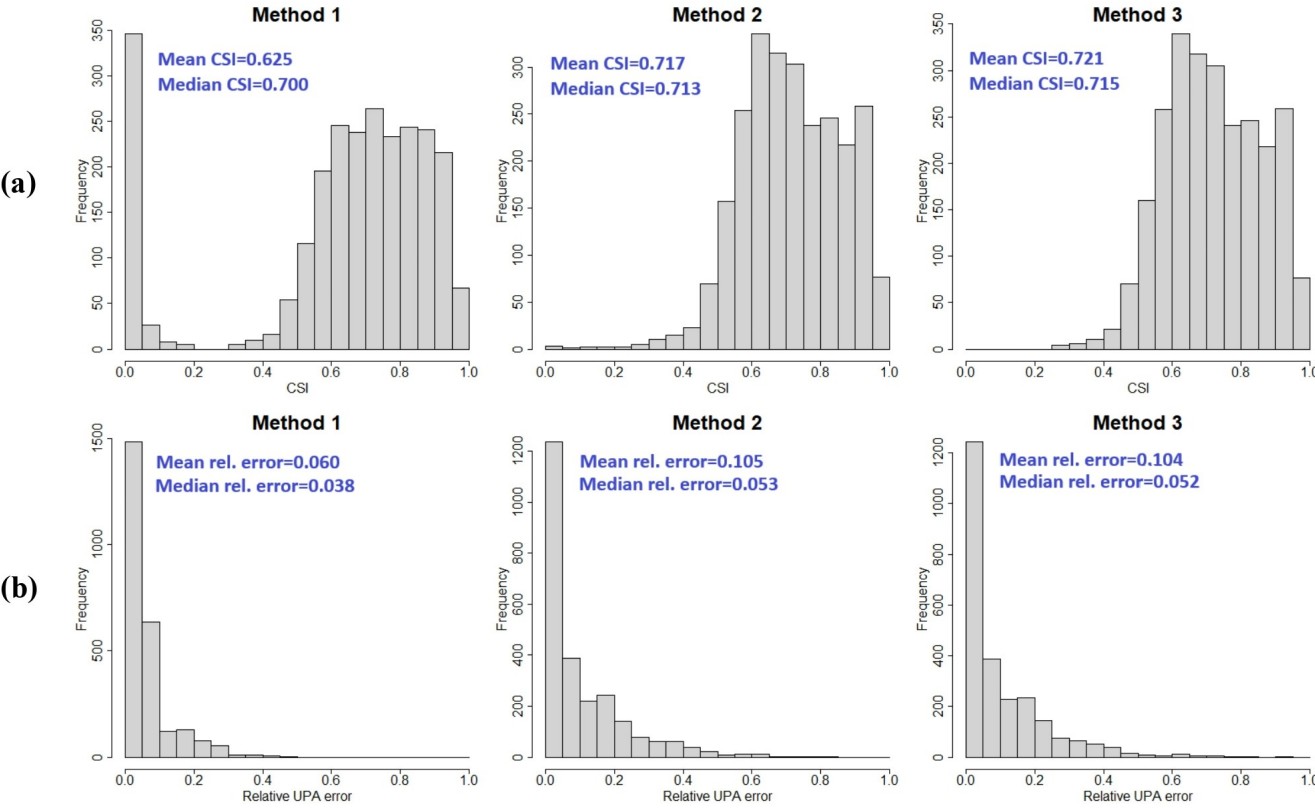

**Figure 7.** Histograms of (a) CSI values and (b) UPA relative error for the three allocation methods.

cannot represent faithfully the river network, and all candidate cells would give a mediocre allocation result. However we argue that Method 3 will give the most correct allocations in these situations. Furthermore, Figure 8 reveals that Method 3 effectively corrects the allocation errors made by Method 1 (indicated by orange circles in the figure). However, it also indicates that there are surprisingly cases where Method 1 yields slightly better CSI scores than Method 3 (nine cases circled in blue). This suggests that the minimum CSI values defined in section 2.3 for the two possible steps of Method 3 may not be optimal, a topic that will be further discussed in Section 4.4.

Finally, the difference in UPAs between each reference catchment and its corresponding coarse resolution catchment was also calculated, even though we decided not to use this metric for the comparison of the three methods, in agreement with the many hydrologists (e.g. Davies and Bell, 2008) who have pointed out that an evaluation based on UPA comparison is highly uncertain. This thesis is supported by the results presented on figure 7b, which show a slightly smaller difference in upstream area for Method 1 than for Methods 2 and 3. Thus, comparing the three allocation methods based on this criterion only would be misleading, because it would not account for all the cases where a river point is allocated to a hydrological modelling cell

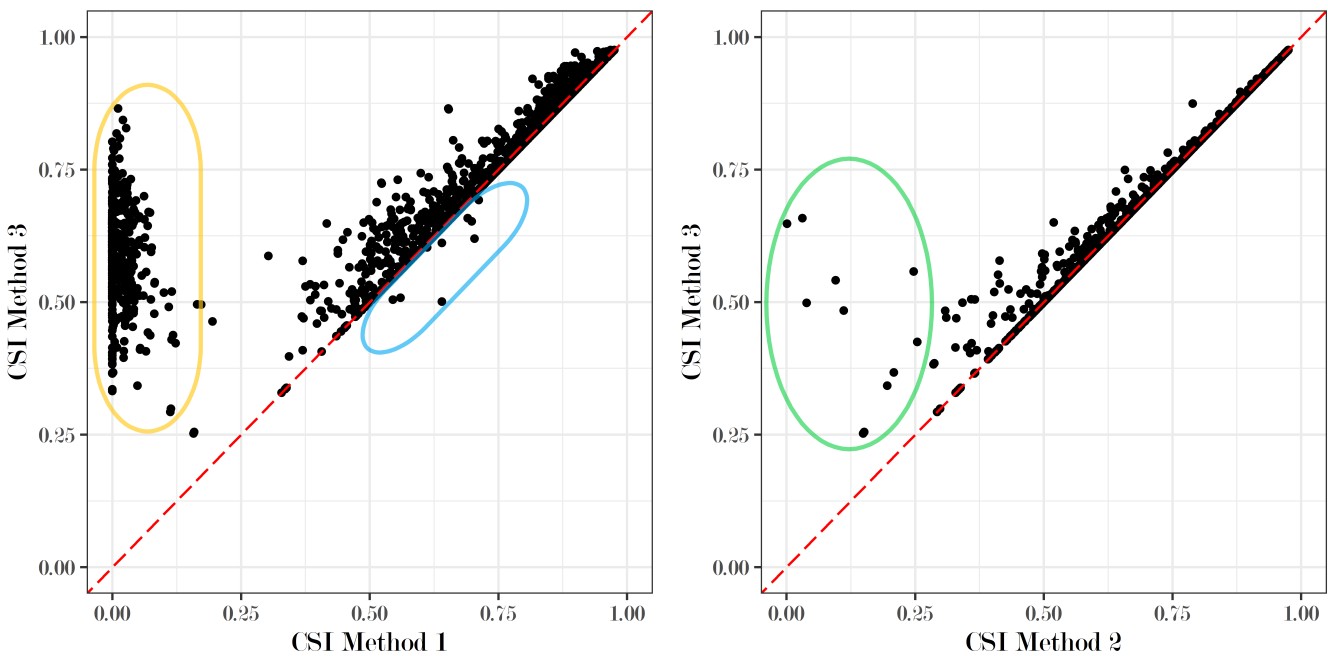

**Figure 8.** Comparison of CSI scores for each outlet, between Methods 3 and 1, and between Methods 3 and 2.

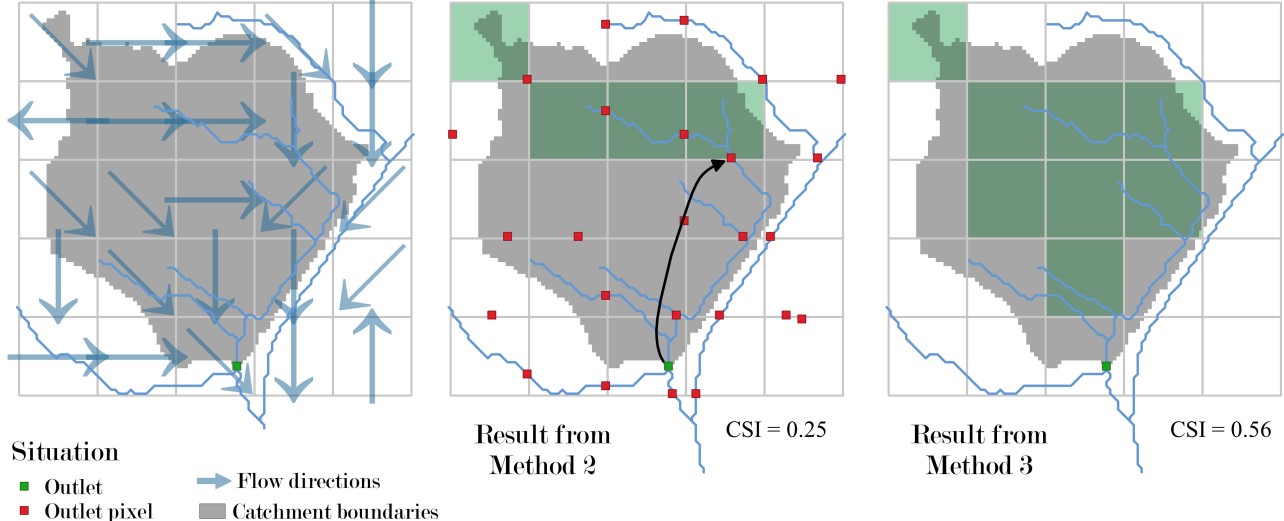

**Figure 9.** An example of high CSI differential between Methods 2 and 3 (basin area $12km^2$). For the sake of comprehension, small ($<5km^2$) tributaries were added to the vectorial network in this figure, explaining why the river outlet can be allocated with Method 2 despite being apparently located between two confluences within the same grid cell.

describing a different upstream catchment, if they have similar UPAs. However, these results show that the relative difference in UPAs remain limited (mostly lower than 15%) for all three methods.

## 4.3 The influence of the upstream basin area

The analysis reveals that basin area plays a significant role in explaining the largest allocation errors. Specifically, among all the river points allocated using Method 1 and having a CSI lower than 0.05, 80% have a basin area smaller than $9km^2$. Similarly, with Method 3, among all the river points with CSI scores lower than 0.6, 100% have a basin area lower than $25km^2$, and 92% have a basin area lower than $10km^2$.

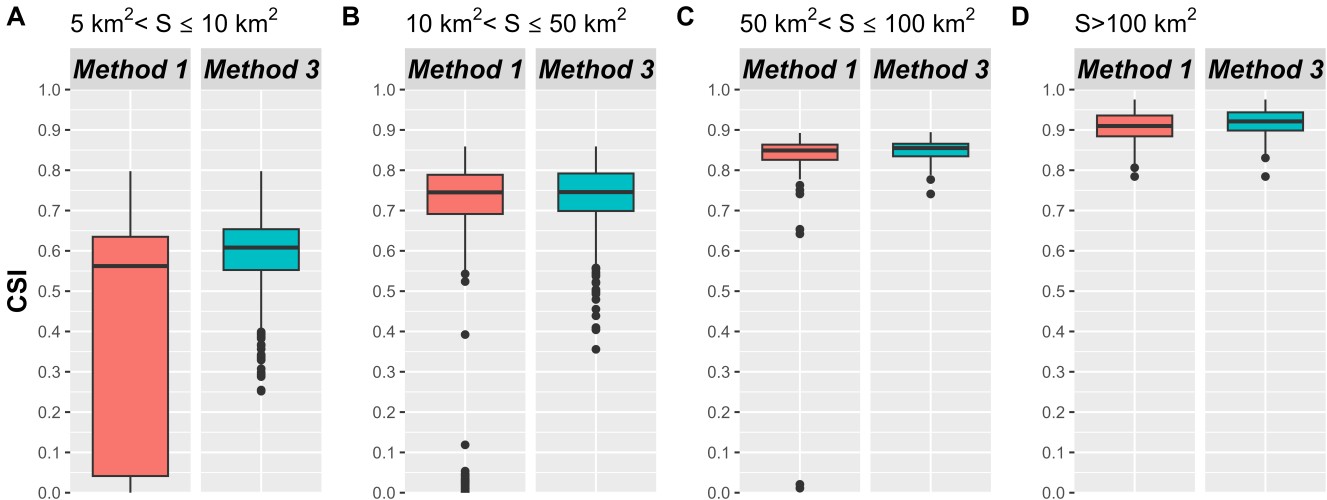

**Figure 10.** Boxplots of CSIs divided into classes of surfaces for Methods 1 and 3.

The boxplot comparisons in Figure 10 highlights two important observations. Firstly, Method 3 effectively prevents the largest allocation errors generated by Method 1. Secondly, these allocation errors predominantly occur for small catchment areas. For catchments larger than $100km^2$, CSI scores systematically exceed 0.75 for both methods. Consequently, Method 1 can be considered a reliable method when applied to the main river network, which explains its widespread usage in previous studies : i.e. for almost all the studied watersheds, the minimum catchment size in various previous research works is between $100$ and $10,000 \ km^2$ (Fekete et al., 2002; Döll and Lehner, 2002; Sutanudjaja et al., 2018; Wang et al., 2018; Zhao et al., 2017; Burek et al., 2020; Polcher et al., 2022).

In addition, Figure 10 also confirms that, with Method 3, the lowest CSI scores are obtained for small catchments - the minimum CSI value is consistently higher than 0.7 for catchment sizes larger than $50km^2$. The low CSI values reflect the uncertainties affecting the boundaries of small basins defined on a 1km resolution grid, particularly pronounced for narrow watersheds. However, it is important to note that the impact of low CSI scores on the representation of rainfall is relatively

minor for small catchments compared to larger ones, because of the limited variability of rainfall at scales of the order of a square kilometre.

## 4.4    Effect of the criterion conditioning Method 3 iteration

While Method 3 appears, without surprise, to be the most reliable method, it has also some shortcomings. One is the limited search zone on the coarse-resolution hydrological grid. The current implementation first considers only the nine surrounding
pixels and then extends the search area to forty-nine surrounding pixels under certain conditions. This approach was chosen to reduce the computation time, which is significantly longer for Method 3 compared to the other two methods. The criterion used for the second iteration depends on the catchment size, based on the assumption that the upstream drainage area (UPA) would influence the allocation results.

    The second iteration of Method 3, is only activated for 70 outlets out of 2580. Extending the research area in the second
iteration improved the allocation for only 10 out of these 70 outlets. However, the increase in CSI for these improved allocations was quite significant, with a median increase of 133%. Moreover, it is obvious on figure 8 that Method 3 does not lead to the optimal allocation in some rare cases : see the 9 cases the circled in blue. A visual analysis indicates that the search area composed of the 49 surrounding grid cells would have solved the problem, if the second iteration of Method 3 would have been activated. Increasing the threshold values for this activation to 0.55 (resp. 0.65) for catchment areas lower (resp. higher)
than 10 $km^2$, would solve the problem encountered for these 9 points in the present case study, but at the cost of higher computing times : 353 points processed in the second iteration of method 3, instead of 70 for the initial theshold values.

## 5    Discussion and conclusions

This comparative study of methods allocating river points to coarse grid cells was driven by the shift in approach from area-based methods to contour-based methods. In this work, we compared these two categories of methods and introduced a new
approach based on topological proximity.

## 5.1    Application domain of each method

The study results revealed that contour-based methods were more relevant and satisfying from a hydrological point of view, although costly in terms of computing time. The introduced topology-based method is a good compromise because it leads to similar allocation quality than the contour-based method. However, it is inoperable when the fine and coarse resolutions river
networks come from different data sources, since it requires the definition of "cells outlet points" as well as the vector-based description of the river network. Moreover, it cannot allocate as many points as the contour-based method.

    The area-based method generated numerous allocation errors, which the contour-based method was able to address for a significant portion of them. However, upon closer examination, it was observed that the performance gap between both methods was more pronounced for small catchments, while being less significant for larger catchments (with $S > 100 km^2$).
The area-based methods thus lead to satisfying results if we only consider river points with large UPAs compared to the grid

cell resolution. Based on the results obtained, we would recommend a minimum factor of 100 between a river point's UPA and the resolution of the hydrological modelling grid for the application of an area-based method.

The transferability of the results outside the test area and to coarser resolutions (i.e. global hydrological models) is debatable, as there are many uncertainties and non-linearities in the representation of hydrological information at such larger scales and coarser resolutions. However, it is very likely that, with coarser resolution grids, allocation problems will increase and that errors related to area-based methods will impact larger catchments (i.e. larger than $100km^2$). Even if this needs to be verified, the "contour-based" method will certainly remain more effective at coarser resolutions than the "area-based" method.

## 5.2 Remaining limitations of allocating river points to coarse grid cells

Even if allocation errors are reduced, some low CSI scores remain with contour-based methods, due to the inherent difficulty of representing the boundaries of small basins at a 1km resolution. The situation illustrated in figure 9 is an example of configurations where different branches of the river system are crossing the same coarse grid cell, which makes it impossible to correctly represent the river network and basin contour with coarse flow directions. However, it is important to note that these inevitable errors resulting in low CSI scores are generally found on small catchments, and are less problematic than low CSI scores on larger catchments, because of the more limited variability of rainfall at smaller spatial scales.

A possible way to reduce these remaining errors could be to allocate river points to multiple grid cells, by either taking the sum of the upstream cells, or the difference of the downstream cells, even though it could complicate the hydrological modelling at a later stage. Another approach that could circumvent the challenges faced would be to use hydrological models structured based on vectorial objects instead of regular grids. These models preserve the topology of river networks and allow seamless integration of observational data. However, vector-based modeling also introduces its own challenges related to data and computational requirements and the need for accurate input data.

*Code and data availability.* All data and codes are provided in an open-access format on the French public data platform Data gouv: https://doi.org/10.57745/7GCCUN (Godet, 2023)

*Author contributions.* JG performed the computational work, helped by PN, and did most of the writing. PN, OP, EG and PJ supervised this work and reviewed the paper. JG, OP and EG replied to the reviewers' comments

*Competing interests.* The authors declare that they have no conflict of interest.

*Acknowledgements.* The BNBV database was provided by the SCHAPI (Service central d'hydrométéorologie et d'appui à la prévision des inondations). The authors would like to thank Patrick Arnaud from INRAE Aix-en-Provence, who wrote and shared the core code used for performing the area-based method, and Nouhaila Mesbahi, who first worked on this method during her internship at Gustave Eiffel University in 2021.

*Financial support.* This research was performed within the framework of the MUFFINS project (ANR-21-CE04-0021-01).

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
