# Peer review of "Technical note: Comparing three different methods for allocating river points to coarse-resolution hydrological modelling grid cells"

_Hydrology and Earth System Sciences, 2023_

## Author Comment (AC1)

**Authors' Response to Reviews of**

**Technical note: Comparing three different methods for allocating river points to coarse-resolution hydrological modelling grid cells**

Juliette Godet, Eric Gaume, Pierre Javelle, Pierre Nicolle, and Olivier Payrastre
*Hydrology and Earth System Science,* `https://doi.org/10.5194/hess-2023-165`
* * *
**RC:** *Reviewers' Comment*,    AR: Authors' Response,    ☐ Manuscript Text

**1. Reviewer #1**

**1.1. General comments**

**RC:** ***The paper investigates different method to allocate locations of stream gauges to the correct river cell in course resolution distributed hydrological models. Three different methods are investigated and compared for the French southeastern Mediterranean region. The methods are based on 1) upstream area and distance; 2) high-resolution river topology; and 3) catchment contour. The methods are compared based on the overlap between the high resolution catchment contour of the gauge and the low resolution catchment contour of the model upstream from the allocated river cell. The topic is relevant and often overlooked. The paper is also generally well written and the methods are mostly well described. However, I have some concerns about the methods and the results as outlined below. I therefore recommend major revisions of the paper.***

AR: First of all, we would like to thank Reviewer #1 for the careful read of our manuscript, and for the emphasis they have placed on understanding each method. We provide point to point answers below, including details about the corresponding modifications of the manuscript.

**1.2. Main comments**

**RC:** ***For gauges which are located between two confluences within one cell, see e.g. P3 in Figure 3, the authors state that these cannot be allocated to the correct river cell using method 2, but can be allocated using method 1 and 3. In my opinion, the only correct allocation would be to both upstream cells (e.g., cells C3 and C5), by comparing the sum of the model discharge against the observed discharge. With method 1 and 3, while the method does assign the gauge to a single cell, I think that is an incorrect allocation for these cases. This is not discussed in the paper. Also, with a small extension, method 2 would actually be able to correctly allocate the gauge to both cells.***

AR: We would rather speak of imperfect allocations than of incorrect allocations for all methods which is an inevitable consequence of the limited spatial resolution provided by the grid cells. In case of figure 3, methods 1 and 3 will probably allocate point P3 to one of the grid cells C3, C5 or C6 depending on the shape of the upstream river network which is not illustrated on the figure. The case of point P3 corresponds to 2% of all the river points to be allocated in the considered region. Method 2 was thought as a simple allocation method consisting in going up and down along the river network, and it did not appear necessary to look for cells outlet points upstream confluences. To discuss these choices in the paper, we suggest to add the following text in Section 4.1, after Table 1:

> It is also important to note that river points similar to P3 in Figure 3, i.e. points located between two confluences within the same grid cell, are not allocated with Method 2. These points represent about 2% of the total number of points to be allocated in the considered region. One potential solution could consist in selecting, among the closest upstream or downstream outlets of grids, the one with the nearest usptream watershed area. Another solution would be to allocate P3 to both upstream cells (C3 and C5 in Figure 3), however in this paper we chose not to permit the allocation to several grid cells. In these cases, Method 2 detects that all allocations to a single grid cell will be imperfect, which is why it was not deemed essential to propose an allocation solution for these specific points, for the sake of comparison.

We also suggest to replace the second paragraph in section 4.2 with:

> In its state, Method 2 cannot allocate 100% of the river outlets, even though it could with a small extension (see previous section). Method 3 was able to allocate the excluded outlets successfully, with a mean CSI of 0.7. Additionally, Figure 7 highlights that Method 3 consistently yields a minimum CSI of around 0.25, whereas Method 2 falls below 0.05 (with 12 river points having CSI < 0.25). Figure 8 provides a more detailed comparison of CSI scores between Method 2 and Method 3. It clearly demonstrates the consistent superiority of Method 3 over Method 2. ...

**RC:** *The authors compare the different methods based on the CSI of overlapping catchment contours, which is also optimized in the allocation process of method 3. I find this single metric for benchmarking the different methods too limited. For a fair comparison, it would be better to use multiple metrics including difference in upstream area (which is also easier to understand). Or if possible, use manually allocated gauges as a reference, to understand the true errors made by each method.*

AR: The manual allocation would indeed serve as a good reference to understand the true errors made by each method, however it would be a very time-consuming for the 2580 river points of the study case and manual allocations are also deemed to errors... However, we argue that the difference in upstream areas (UPA) metric can be misleading, as explained on figure 1 (in the article), and should be considered with caution. We also maintain that, in this work, the addressed problem is the correct delineation of the basin contours, rather than the correct value of upstream watershed area.

We have nonetheless calculated the difference in UPA relative to each method, and the results are presented on figure 1.

[Figure]

Figure 1: Results of the calculation of difference in UPAs

As expected, Method 1 provides the best results according to this metric. The problem is that it doesn't account for all the cases where a river point is allocated to a hydrological modelling cell describing a different upstream catchment, if they have similar UPAs. We suggest to add this figure in the manuscript to support the fact that an evaluation based on UPA comparison can be misleading. We therefore suggest to add the following text at the end of section 4.2:

> Finally, the difference in UPAs between each reference catchment and its corresponding coarse resolution catchment was also calculated, even though we decided not to use this metric for the comparison of the three methods, in agreement with the many hydrologists (e.g. Davies and Bell, 2008) who have pointed out that an evaluation based on UPA comparison is highly uncertain. This thesis is supported by the results presented on figure 7b, which show a slightly smaller difference in upstream area for Method 1 than for Methods 2 and 3. Thus, comparing the three allocation methods based on this criterion only would be misleading, because it would not account for all the cases where a river point is allocated to a hydrological modelling cell describing a different upstream catchment, if they have similar UPAs. However, these results show that the relative difference in UPAs remain limited (mostly lower than 15%) for all three methods.

[Figure]

Figure 7: Histograms of (a) CSI values and (b) UPA relative error for the three allocation methods

**1.3. Minor comments**

**RC:** *It would be helpful to illustrate in Figure 2-4 to which river cell the gauges are allocated.*

AR: This is already indicated in figure 3 (see Allocation process). However, in the revised manuscript, the allocated grid cells will be indicated in the captions of Figures 2 and 4.

**RC:** *Line 89: Consider using a more commonly used notation for CSI (see e.g., Fleischmann et al., 2019). It is also not entirely clear to me how the CSI is calculated because of the different resolutions of the catchment contours. Is the CSI calculated based on the low resolution catchment contour of the model or the high resolution catchment contour of the gauge? This could make quite a difference for certain catchments.*

**AR:** As noted in line 94, the CSI is sometimes known as Figure of Merit, Intersection over Union Index, and is also referred as Fit metric as in Fleischmann et al., 2019 (this will be added in the text). All these scores are scrictly identical, and to our knowledge, the Critical Success Index remains the most generic term used in the litterature for this metric, which is widely used when dealing with contingency tables. We thus suggest to modify line 94 as following:

> It can be noted that the CSI has often been used with alternative denominations in previous studies, such as the Intersection over Union criterion (Munier and Decharme, 2022; Burek and Smilovic, 2022), the Figure of Merit (Li and Wong, 2010), or Fit Metric (Fleishcmann et al., 2019).

The CSI is calculated here based on the high resolution catchment contour of the river point. In order to clarify this point, we suggest to add the following text line 104:

> ... and will be used hereafter. The CSI is calculated based on the low resolution catchment contours (reference contours).

**RC:** *Line 94: The inline formula is hard to read and the variables unclear as they refer to criteria used in other papers. Could the authors explain the variables shortly here to make interpretation easier?*

**AR:** We suggest to remove the formula, and only keep the text as written in the previous answer.

**RC:** *Figure 9: Can you add the outflow points of all cells in Figure 9B to better understand why the cell just upstream from the gauge is not found?*

**AR:** We have added the outflow points of all cells in Figure 9B. We have also drawn small tributaries that did not initally appear because their upstream area is inferior to $5km^2$. However here they help understand why the cell just upstream from the gauge is not an option (its outflow point represents the very small tributary, which has a smaller upstream area but occupies more space in the cell than the main river reach). We suggest to replace Figure 9 by the following:

[Figure]

Figure 9: An example of high CSI differential between Methods 2 and 3 (basin area $12km^2$)

**RC:** *Figure 10: the stacked histograms are difficult to read. Consider using a different histogram style.*

 **AR:** We suggest to use boxplots instead of histograms, and to reduce the number of surface classes to make the figure less busy.

[Figure]

Figure 10: Boxplots of CSIs divided into classes of surfaces for Methods 1 and 3

**RC:** *Line 211: It is stated that method 2 requires a vector-based description of the river network (which I guess is the same a the high resolution river topology / flow directions?). However, if I understand correctly, method 3, would require a vector-based description of the catchment contour which is not mentioned here.*

 **AR:** Indeed, in both case, vectorial data (high resolution river topology or high resolution catchment contours) is needed. We thus suggest to remove "as well as the vector based description of the river network" from the text.

**RC:** *Line 222: I suggest to mention vector-based models already in the introduction to emphasize that issue and proposed methods are specific to raster-based models.*

 **AR:** We suggest to add the following text after the first sentence of the introduction:

> ...or evaluation purposes. Vector-based hydrological models are adequate to meet these objectives, because it is straightforward to locate a gauging station along the river network. However, when using gridded models...

---

## Author Comment (AC2)

**Authors' Response to Reviews of**

**Technical note: Comparing three different methods for allocating river points to coarse-resolution hydrological modelling grid cells**

Juliette Godet, Eric Gaume, Pierre Javelle, Pierre Nicolle, and Olivier Payrastre
*Hydrology and Earth System Science,* `https://doi.org/10.5194/hess-2023-165`

RC: *Reviewers' Comment*,     AR: Authors' Response,     ☐ Manuscript Text

**2. Reviewer #2**

**2.1. General comments**

RC: *Godet et al. provide comparison of methods to allocate river points to the most appropriate hydrological model grids. This task is important and becoming more important given the rise in number of gridded hydrological models being made available within hydrological research and operations. The paper compares three allocation methods: area-based, topology-based, and contour-based. The results indicate that contour-based methods, though computationally expensive, are more hydrologically relevant, with topology-based methods serving as a reasonable compromise. Area-based methods lead to numerous allocation errors, particularly for small catchments, and are recommended only for river points with large upstream drainage areas compared to the grid cell resolution.*

*I do have some questions for the authors about the transferability of their results outside the current test area in the Eastern Mediterranean region of France covering an area of 15,000 km2 with the largest catchment size considered only 3000 km2. They define "coarse-resolution" as a 1km hydrological modelling grid size. In the context of global hydrological modelling, 1 km is the benchmark to be deemed "hyperresolution" (Wood et al., 2011). While this paper is over a decade old, there remains relatively few hydrological models running at 1km scale, even at national scales. For a user of a model running at 5km or 10km or even coarser, are the conclusions in Godet et al. still valid? What about transferability to other regions? We know hydrology is heterogenous with complex river networks such as braided rivers; we know that high quality DEMS/vector river networks are not available in all regions, and that the quality of upstream catchment size metadata information can be missing or uncertain in some regions of the world. Very few of these uncertainties are considered or at least discussed in the paper.*

*There has been a limited amount of research comparing difference approaches to this important technical issue, therefore the paper by Godet et al. is a very useful reference to help guide others on selecting the most appropriate method/understanding the limitations of simpler methods. However, at a minimum I suggest more effort is needed to discuss uncertainties and transferability outside the limited test case used. I recommend this paper for publishing in HESS after such changes are made.*

AR: We would like to sincerely thank Reviewer#2 for their comments and for highlighting the lack of discussions about transferability, which we will take into account in the revised manuscript to improve the quality of the paper. We provide below detailed answers showing how we plan to adapt the manuscript according to these suggestions.

**2.2. Main comments**

**RC:** *Pg3 L46-48: As per my summary above, from the perspective of gridded hydrological models that are not run at very local scales, then the definition of "coarse-resolution hydrological grid (1km×1km)" could arguably be considered "high resolution". 1 km is the benchmark to be deemed "hyperresolution" (Wood et al., 2011) for global scale models. To what extent are these conclusions/methods transferable to coarser model resolutions that are often used (e.g. 5km, 10km or coarser)? Perhaps qualifying why 1km is deemed "coarse" and if so, does this limit the transferability of methods/conclusions?*

**AR:** In this case, the hydrological model is intended for the regional scale, we could even imagine to implement it for the fine resolution (50m). However, the same problem could arise for hydrological modelling applied on a continental scale where the resolution will be coarser than 1km (i.e. 5 to 10 km), whereas the DTMs available worldwide have a resolution of a few tens of metres (e.g. 90m for SRTM). In that case, allocation problems will probably be even more complex, with higher risk of errors. Even if that needs to be verified, it is likely that the errors related to area-based methods will concern larger catchments (i.e. larger than $100 km^2$). To make this discussion appear in the manuscript, we suggest to add the following text in the introduction (line 52):

> In this study, $1km \times 1km$ is considered as "coarse" resolution because the hydrological model is intended for the regional scale. However, the same problem could arise for hydrological modelling applied on a continental scale where the resolution will be coarser than 1km (i.e. 5 to 10 km).

And the following text in the conclusion (line 217):

> This recommendation is valid for the tested resolutions, however, as indicated in the introduction, the problem raised in this paper will be encountered also for coarser resolutions, especially when using global hydrological models. The transferability of the results outside the test area is debatable, as there are many uncertainties and non-linearities in the representation of hydrological information at larger scales. However, it is very likely that, with coarser resolution grids, allocation problems will increase and that errors related to area-based methods will impact larger catchments (i.e. larger than $100 km^2$). Even if that needs to be verified, the "contour-based" method will certainly remain more effective at coarser resolutions than the "area-based" method.

**RC:** *Pg4, L71: The parameter R (here R < 3) seems to be very dependent on the model resolution and catchment size. Why was R < 3 selected and was a sensitivity done on its selection? How would varying R to be larger or smaller impact the results? It will also be depended on catchment area of the station that you are trying to allocate. For example, for a catchment area of < 3000 km2 (as is considered here) then R < 3 might be appropriate. However, if you are mapping river gauges in global gridded models and are considering stations in downstream sections of major world river basins (e.g. Amazon, Danube, etc.), then you would need an R much larger than 3 – this parameter needs to vary by both grid resolution and catchment size.*

**AR:** R<3 has indeed been chosen as a result of a sensibility analysis. It was found that R<3 was a good compromise between too large a radius, which increases the risk of error, and too small a radius, which risks not searching far enough for candidates, for the study area. This choice is rarely justified in other works and it does depend on both model resolution and catchment size. We suggest to add, the following text at the end of section 2.1, line 72:

> ... and distance criteria. Also, if a maximum difference between UPAs of 30% is a recurrent choice in the literature (e.g Burek et al., 2020) regardless of the studied model resolutions, the distance criterium R<3 is more study-dependant. In the present study, it appeared after some tests as a good compromise providing accurate results with reasonable computation times. However when using global-scale hydrological models and coarser grids, the value of R may have to be adjusted.

**RC:** *Pg4, L74-79: How would method 2: 'topology-based method' work when the underlaying gridded hydrological model river network is different to the vector network. For example, if there are spatial mismatches where the vector for a river section does not overlap with the most appropriate model cell? Often the data source to derive a hydrological model river network grid is different from a vector river network.*

**AR:** It would be impossible to use Method 2 in that case, because this topolgy-based method requires the notion of "cells' outlet points", thus it needs consistancy between the hydrological grid and the vector network. This is indeed a major drawback of the method. We propose to add the following text in section 4.1, line 141:

> ... as it relies on the IHU upscaling. As a consequence, it makes this method inoperable in contexts where the coarse resolution gridded network does not coincide with the river network (i.e. both networks may come from different data sources).

**2.3. Technical comments**

**RC:** *P6, L98-99: Can you please justify use of CSI = 0.4 and 0.6, and are these applicable to much larger catchment sizes?*

**AR:** Ideally, only the threshold CSI=0.6 should be used to ensure the quality of the allocation process, however it would be unrealistic for catchments which sizes are close to the pixel size $1km^2$. The threshold CSI=0.4 enables to go looking for further cells even for small catchments. As explained in section 4.4, these thresholds should be adjusted according to the users' needs. When using global-scale hydrological models, it is the threshold of $10km^2$ that will need adjusting. We propose to add the following indication line 99:

> The CSI thresholds can be adjusted (see section 4.4), as well as the surface threshold which depends on the model resolution.

**RC:** *Pg 6, L99: "the search area is extended to the 49 closest grid cells": why 49, please elaborate?*

**AR:** In a second iteration, we aim at extending our research area (1: $3 \times 3 = 9$ surrounding pixels, 2: $7 \times 7 = 49$ surrounding pixels). As explained in lines 196-197, increasing the research area in the second iteration above than the 49 surrounding cells does not change much the results. However, as mentioned before, this will depend on the model resolution. We propose to add the following text line 99:

> ...the search area is extended to the $7 \times 7 = 49$ closest grid cells. The CSI thresholds can be adjusted (see section 4.4), as well as the surface threshold and the extended research area, since they may depend on the model resolution.

---

## Author Response (AR2)

**Authors' Response to Second Reviews of**

**Technical note: Comparing three different methods for allocating river points to coarse-resolution hydrological modelling grid cells**

Juliette Godet, Eric Gaume, Pierre Javelle, Pierre Nicolle, and Olivier Payrastre
*Hydrology and Earth System Science,* `https://doi.org/10.5194/hess-2023-165`
* * *
**RC:** *Reviewers' Comment*,     AR: Authors' Response,     ☐ Manuscript Text

**1. Reviewer #1**

**1.1. General comments**

**RC:** *The authors have improved the manuscript on most of the points raised. However, I think it would still benefit from discussing limitations of the proposed method more generally and therefore I suggest minor revisions before publication.*

AR: First of all, we would like to thank Reviewer #1 for accepting to review our article a second time, and for the very useful comments they provided each time. We will explain hereafter how we plan to adapt the manuscript according to their recommendations.

**1.2. Main comments**

**RC:** *I suggest the authors to discuss the limitations of allocating gauges to coarse resolution gridded flow directions more generally, possibly in a separate section. The issue highlighted for method 2 also applies to method 1 and 3 and is inherent to coarse resolution grid based representation of river networks. Assigning gauges to multiple cells could limit the errors made, i.e. in the cases presented in Figure 3 and 9 the most correct result is obtained by either taking the sum of the upstream cells or difference of the downstream cells. However, errors will remain especially in cases where a coarse resolution representation of the river network is simply not possible (e.g. Figure 9). The authors are right to point out at the end of the conclusions that in these situations vector based model would circumvent issues introduced by coarse resolution gridded models.*

AR: We do agree that the limitations should appear more clearly and be a little bit more elaborated. To link up with a suggestion from Reviewer #3, we propose to introduce three subsections in the conclusion, and to change the text as follows:
* * *
**5. Discussion and conclusions**

This comparative study of methods allocating river points to coarse grid cells was driven by the shift in approach from area-based methods to contour-based methods. In this work, we compared these two categories of methods and introduced a new approach based on topological proximity.

**5.1 Application domain of each method**
* * *
The study results revealed that contour-based methods were more relevant and satisfying from a hydrological point of view, although costly in terms of computing time. The introduced topology-based method is a good compromise because it leads to similar allocation quality than the contour-based method. However, it is inoperable when the fine and coarse resolutions river networks come from different data sources, since it requires the definition of "cells outlet points" as well as the vector-based description of the river network. Moreover, it cannot allocate an as many points as the contour-based method.

The area-based method generated numerous allocation errors, which the contour-based method was able to address for a significant portion of them. However, upon closer examination, it was observed that the performance gap between both methods was more pronounced for small catchments, while being less significant for larger catchments (with $S > 100km^2$). The area-based methods thus lead to satisfying results if we only consider river points with large UPAs compared to the grid cell resolution. Based on the results obtained, we would recommand a minimum factor of 100 between a river point's UPA and the resolution of the hydrological modelling grid for the application of an area-based method.

The transferability of the results outside the test area and to coarser resolutions (i.e. global hydrological models) is debatable, as there are many uncertainties and non-linearities in the representation of hydrological information at such larger scales and coarser resolutions. However, it is very likely that, with coarser resolution grids, allocation problems will increase and that errors related to area-based methods will impact larger catchments (i.e. larger than $100km^2$). Even if this needs to be verified, the "contour-based" method will certainly remain more effective at coarser resolutions than the "area-based" method.

**5.2 Remaining limitations of allocating river points to coarse grid cells**

Even if allocation errors are reduced, some low CSI scores remain with contour-based methods, due to the inherent difficulty of representing the boundaries of small basins at a 1km resolution. The situation illustrated in figure 9 is an example of configurations where different branches of the river system are crossing the same coarse grid cell, which makes it impossible to correctly represent the river network and basin contour with coarse flow directions. However, it is important to note that these inevitable errors resulting in low CSI scores are generally found on small catchments, and are less problematic than low CSI scores on larger catchments, because of the more limited variability of rainfall at smaller spatial scales.

A possible way to reduce these remaining errors could be to allocate river points to multiple grid cells, by either taking the sum of the upstream cells, or the difference of the downstream cells, even though it could complicate the hydrological modelling at a later stage. Another approach that could circumvent the challenges faced would be to use hydrological models structured based on vectorial objects instead of regular grids. These models preserve the topology of river networks and allow seamless integration of observational data. However, vector-based modeling also introduces its own challenges related to data and computational requirements and the need for accurate input data.

**1.3. Minor comments**

**RC:** *Figure 2 and 4. The caption reads "In this situation, the river point is allocated to the [x,y] cell in the grid (zero-based numbering)." If possible I suggest to add row and column numbers or at least state row x and column y as the current notation is not clearly defined.*

AR: According to the recommendations of Reviewer #3, we have indicated the chosen cell with hatching on Figure

2, therefore removing the row/columns notation. On Figure 4, we suggest the following changes:

[Figure]

Figure 4: Illustration of Method 3: CSI calculations for the nine candidate cells. In this situation, the river point is allocated to the candidate cell of the blue box scenario, which maximises the CSI (CSI = 0.80).

**RC:** *Figure 9. It seems that in this situation the flow directions on the coarse resolution cannot represent the full river network, as two rivers run parallel trough the a single cell (i.e. the cell on the upper right of the gauge). In this situation upscaled flow directions are basically inconsistent with the outlet locations. The most correct allocation would come from the difference between the two downstream cells. I suggest to add this to the discussion of this figure.*

AR: Thank you for noticing this. This issue with two rivers crossing the same cell are often at the roots of such situations, and it makes it impossible from the very start to find a correct grid cell for the allocation, though it is argued that Method 3 limits the damage in these rare situations. We propose to add the following text line 184:

> ... resulting in notable differences in UPAs. Figure 9 shows an example of these situations, which are often complex configurations where different branches of the network are in close proximity to each other, often within the same grid cell. As a consequence, the upscaled flow directions cannot represent faithfully the river network, and all candidate cells would give a mediocre allocation result. However we argue that Method 3 will give the most correct allocations in these situations.

**2. Reviewer #3**

**2.1. General comments**

**RC:** *This technical note focuses on comparing three methods for allocating river points to coarser-resolution grids for hydrological modelling. The authors describe in detail the methods and evaluate them on a large area in South-eastern France. The work provides useful insights on a relevant question and therefore I believe it could be useful for the hydrological modelling community.*

**AR:** We would like to thank Reviewer #3 for accepting to review our manuscript and for their positive comments. We provide below detailed answers showing how we plan to adapt the manuscript according to their suggestions.

**2.2. Minor comments**

**RC:** *Figure 2: Can you identify the coarse-grid cell in the drawing (e.g. with hatching)?*

**AR:** We agree that identifying the chosen coarse-grid cell in the figure would be better than in the legend. We adapt the figure as follows:

[Figure]

Figure 2: Illustration of Method 1: grid cells candidates for a specific river point. In this situation, the river point is allocated to the green-hatched grid cell.

**RC:**   *Lines 110-114: Wouldn't make sense to extend the search area first within 5x5=25 closest cells, and then within 7x7 cells?*

**AR:**   That was our first idea too, however extending the research area from 9 to 25 cells did not correct most of the targeted cases, which had to be sought even further. That is why we directly extended to the 49 closest grid cells. We suggest to add the following sentence at line 108:

> ... is extended to the $7 \times 7 = 49$ closest grid cells. Tests carried out prior to this choice showed that extending the research area to the $5 \times 5 = 25$ closest grid cells did not allow most of the targeted cases to be corrected.

**RC:**   *Table 1: I assume that these computational times refer to the test area. Maybe the authors could say something about how run times could increase on larger areas. On this point, my opinion is that applying a method with long computational times might be acceptable if resulting accuracy is significantly higher than for other methods, given that river point allocation is not done routinely.*

**AR:**   We agree. We suggest to change lines 150-151 as follows:

> Method 1 and 2 demonstrate superior performance in terms of computation times. The most time-consuming stage of Method 3 is the computation of the upstream catchment for each candidate cell, and its comparison with the reference catchment. The duration of these processes will naturally increase with the size of the considered case study and the number of points to be allocated. However, the computation times might not be considered as a crucial issue, since the allocation of river points to grid cells need do be performed only once, before hydrological modelling. Method replicability is probably a more important issue for this work. Method 2 is limited...

**RC:**   *Section 4.2: I am wondering how these results could change with a different coarse-grid resolution. Perhaps the authors could briefly elaborate on that*

**AR:**   We argue in the conclusion that our results are valid for the tested resolutions, but we believe that with coarser resolution grids, allocation problems will increase and that errors related to area-based methods will probably impact larger catchments. We also believe, as it is written in the conclusion, that contour-based methods would remain more effective than area-based methods for coarser resolutions. These discussions might not be very visible, thus we have suggested, based on the recommendations of Reviewer #1, to put them in a separate section (see the answer to reviewer #1 here).